# High Concentration or Combined Treatment of Antisense Oligonucleotides for Spinal Muscular Atrophy Perturbed *SMN2* Splicing in Patient Fibroblasts

**DOI:** 10.3390/genes13040685

**Published:** 2022-04-13

**Authors:** Yogik Onky Silvana Wijaya, Emma Tabe Eko Niba, Hisahide Nishio, Kentaro Okamoto, Hiroyuki Awano, Toshio Saito, Yasuhiro Takeshima, Masakazu Shinohara

**Affiliations:** 1Department of Community Medicine and Social Healthcare Science, Kobe University Graduate School of Medicine, 7-5-1 Kusunoki-cho, Chuo-ku, Kobe 650-0017, Hyogo, Japan; yogik.onky.s@mail.ugm.ac.id (Y.O.S.W.); niba@med.kobe-u.ac.jp (E.T.E.N.); mashino@med.kobe-u.ac.jp (M.S.); 2Department of Biochemistry, Faculty of Medicine, Public Health, and Nursing, Universitas Gadjah Mada, Jalan Farmako, Yogyakarta 55281, Indonesia; 3Department of Occupational Therapy, Faculty of Rehabilitation, Kobe Gakuin University, 518 Arise, Ikawadani-cho, Nishi-ku, Kobe 651-2180, Hyogo, Japan; 4Department of Pediatrics, Ehime Prefectural Imabari Hospital, 4-5-5 Ishii-cho, Imabari, 794-0006, Ehime, Japan; kentaro206@gmail.com; 5Department of Pediatrics, Kobe University Graduate School of Medicine, 7-5-1 Kusunoki-cho, Chuo-ku, Kobe 650-0017, Hyogo, Japan; awahiro@med.kobe-u.ac.jp; 6Department of Neurology, National Hospital Organization Osaka Toneyama Medical Center, 5-1-1 Toneyama, Toyonaka 560-8552, Osaka, Japan; saito.toshio.cq@mail.hosp.go.jp; 7Department of Pediatrics, Hyogo College of Medicine, 1-1 Mukogawa-cho, Nishinomiya 663-8501, Hyogo, Japan; ytake@hyo-med.ac.jp

**Keywords:** spinal muscular atrophy, *SMN1*, *SMN2*, splicing, antisense oligonucleotide, cryptic exon

## Abstract

Spinal muscular atrophy (SMA) is caused by *survival motor neuron 1 SMN1* deletion. The *survival motor neuron 2 (SMN2)* encodes the same protein as *SMN1* does, but it has a splicing defect of exon 7. Some antisense oligonucleotides (ASOs) have been proven to correct this defect. One of these, nusinersen, is effective in SMA-affected infants, but not as much so in advanced-stage patients. Furthermore, the current regimen may exhibit a ceiling effect. To overcome these problems, high-dose ASOs or combined ASOs have been explored. Here, using SMA fibroblasts, we examined the effects of high-concentration ASOs and of combining two ASOs. Three ASOs were examined: one targeting intronic splicing suppressor site N1 (ISS-N1) in intron 7, and two others targeting the 3′ splice site and 5′ region of exon 8. In our experiments on all ASO types, a low or intermediate concentration (50 or 100 nM) showed better splicing efficiency than a high concentration (200 nM). In addition, a high concentration of each ASO created a cryptic exon in exon 6. When a mixture of two different ASOs (100 nM each) was added to the cells, the cryptic exon was included in the mRNA. In conclusion, ASOs at a high concentration or used in combination may show less splicing correction and cryptic exon creation.

## 1. Introduction

Spinal muscular atrophy (SMA) is a common motor neuron disease that is inherited in an autosomal recessive manner [1]. It has a high incidence, affecting 1 in 11,000 newborns [1].

SMA is clinically divided into five groups [1]: type 0 (the most severe form with onset in the prenatal period; severe respiratory problems after birth), type 1 (a severe form with onset before 6 months of age; unable to sit unsupported), type 2 (an intermediate form with onset before 18 months of age; able to sit unaided, but unable to stand or walk), type 3 (a mild form with onset after 18 months of age; able to stand and walk unaided), and type 4 (the mildest form with age of onset from adolescence to adulthood; able to stand and walk unaided). Almost all patients with SMA type 0 die within a few weeks after birth, and most patients with SMA type 1 die or become dependent on a ventilator before reaching 2 years of age [1].

SMA is caused by defects of the *survival motor neuron 1* (*SMN1*) gene located on chromosome 5q13 [2]. More than 90% of SMA patients are homozygous for *SMN1* deletion [2]. Defects of *SMN1* reduce the level of survival motor neuron (SMN) protein, leading to motor neuron dysfunction [3].

A gene homologous to *SMN1* is present at the same locus of chromosome 5q13 [2], which is named the *survival motor neuron 2* (*SMN2*) gene. *SMN2* encodes the same SMN protein as *SMN1* produces. A previous report [2] described that complete loss of *SMN2* was not observed in any SMA patients with homozygous *SMN1* deletion, suggesting that its complete loss results in embryonic lethality [4,5]. Thus, SMA patients with homozygous deletion of *SMN1* have at least one copy of *SMN2*. However, *SMN2* cannot fully compensate for homozygous *SMN1* deletion because *SMN2* produces only a small amount of full-length SMN protein due to the splicing defect of exon 7 [6]. 

It is important to note that, based on the mechanism of *SMN2* exon 7 splicing, “correction of the splicing defect of *SMN2* exon 7” and “modulation of *SMN2* exon 7 splicing” have been explored as strategies for treating SMA [7,8]. Many candidate therapeutic approaches using antisense oligonucleotides (ASOs) have been explored in pursuit of these strategies [9,10,11,12,13,14,15,16,17]. 

Until recently, SMA was incurable, but treatments for this disease are now emerging [18]. For example, the United States Food and Drug Administration (FDA) approved nusinersen (Spinraza®; Biogen Inc., Cambridge, MA, USA) as the first drug for SMA in 2016, onasemnogene abeparvovec (Zolgensma®; AveXis Inc., Bannockburn, IL, USA/Novartis, Basel, Switzerland) as the second drug in 2019, and risdiplam (Evrysdi®; PTC Therapeutics, Inc., South Plainfield, NJ, USA/F. Hoffmann-La Roche Ltd., Basel, Switzerland) as the third drug in 2020. 

These drugs are completely different in terms of their dosage form and route of administration. Nusinersen is an ASO drug that is intrathecally administered every few months [19]. It targets an intronic splicing suppressor site (ISS-N1) in *SMN2* intron 7, corrects the splicing defect of *SMN2* exon 7, and leads to the production of full-length SMN protein [15,16]. Onasemnogene abeparvovec is an adeno-associated virus vector drug carrying *SMN* complementary DNA encoding the missing SMN protein, which is intravenously administered only once [20]. Meanwhile, risdiplam is a small-molecule drug that corrects the splicing defect of *SMN2* exon 7 and leads to production of the full-length SMN protein. It is orally administered every day [21,22,23].

Currently, more than 10,000 patients worldwide are being treated with nusinersen [24]. Nusinersen is effective for improving the motor function of newborns and infants with SMA [25,26], but its efficacy is limited in advanced-stage patients [27]. In addition, a ceiling effect of the current dosage regimen regarding the improvement of motor function has also been reported [28]. To overcome these problems, the effects of nusinersen at a high dose or in combination with different drugs should be explored. 

Concerning treatment with a high dose of nusinersen, a clinical trial of such treatment, the DEVOTE study, is now ongoing (ClinicalTrials.gov Identifier: NCT04089566). However, based on studies using SMA cells, Ottesen et al. suggested that a high concentration of nusinersen might have off-target effects on the expression of certain genes [29]. 

Concerning treatments involving nusinersen in combination with other drugs, Pao et al. reported that dual targeting of both the ISS-N1 site in *SMN2* intron 7 and the 3′ splice site/5′ region of *SMN2* exon 8 by two ASOs increased the level of full-length *SMN* transcription more effectively than a single ASO targeting the ISS-N1 site [30]. However, such dual targeting therapy with two ASOs has not been pursued further. Recently, Harada et al. reported five infants who received nusinersen and onasemnogene abeparvovec [31]. In their study, no adverse effects were noted in the patients, and all patients exhibited improvements in their motor function. Nonetheless, the prolonged use of corticosteroid and careful monitoring of liver toxicity are necessary when onasemnogene abeparvovec therapy is combined with nusinersen therapy [31].

In this study, we examined the effect of a high dose of ASOs and the effect of combining two ASOs. For this purpose, SMA cells were transfected with three ASOs: one targeting the ISS-N1 site in *SMN2* intron 7, and two others targeting the 3′ splice site or the 5′ region of *SMN2* exon 8. Our study clarified that ASOs at a high concentration or in combination may result in adverse effects including reduced correction of the splicing of *SMN2* exon 7 and the activation of cryptic splice sites in *SMN2* intron 6.

## 2. Materials and Methods

### 2.1. Cell Culture and Transfection Protocol 

GM00232 SMA type 1 patient fibroblasts were obtained from Coriell Cell Repositories (Camden, NJ, USA). According to the provider’s information, donor subject had 2 copies of the *SMN2* gene and is homozygous for deletion of exons 7 and 8 of the *SMN1* gene. The cells were cultured in Eagle’s minimum essential medium containing 1% antibiotic and antimycotic solution (consisting of 10,000 U/ml penicillin, 10,000 μg/ml streptomycin, 25 μg/ml amphotericin B) (Nacalai Tesque, Inc., Kyoto, Japan) and 15% fetal bovine serum (Sigma Aldrich, St. Louis, MO, USA), and incubated at 37°C in a 5% CO_2_ humidified atmosphere prior to transfection. We transfected 1 × 10^6^ SMA type 1 fibroblasts in a transfection mixture with a final volume of 10 μL containing ASO and transfection buffer using the Neon transfection system (Invitrogen, Carlsbad, CA, USA). The cells were pulsed twice with a voltage of 1400 V and a width of 20 ms (with each pulse lasting 20 ms). Then, the cells were quickly transferred into 1 mL of medium and incubated for 48 h before harvesting. All plates were pre-coated with StemSure® 0.1% w/v Gelatin Solution (WAKO, Osaka, Japan), in accordance with the manufacturer’s protocol.

### 2.2. Treatments with ASOs

All ASOs containing 2’-*O*-methyl and phosphorothioate backbone modifications (2OMeAO) were purchased from Fasmac Co. Ltd. (Atsugi, Japan). In this study, three ASOs were examined: ASO targeting ISS-N1 (ASO-NUS, named after nusinersen), ASO targeting the sequence within exon 8 (ASO-EX8, named based on the target exon), and ASO targeting the intron 7-exon 8 splice site junction (ASO-SSJ, named based on the 3′ splice-site location) (Figure 1A). The sequences of ASOs were as follows: ASO-NUS, 5′-UCA CUU UCA UAA UGC UGG-3′ [16]; ASO-SSJ, 5′-CUA GUA UUU CCU GCA AAU GAG-3′ [9]; and ASO-EX8, 5′-AUC UUC UAU AAC GCU UCA CAU UCC A-3′ [17].

Our study using the above-mentioned ASOs consisted of two parts. First, to explore the dose effect of ASOs, each ASO was transfected into SMA type 1 fibroblasts at three concentrations (50, 100, and 200 nM), followed by analysis of *SMN2* transcript expression after 48 h of incubation at 37 °C. Second, to explore the effect of combining ASOs, combinations of two ASOs, each at a concentration of 100 nM (total 200 nM), were also transfected into SMA type 1 fibroblasts, after which *SMN2* transcript expression was analyzed following 48 h of incubation at 37 °C. The combinations were ASO-NUS/ASO-EX8, ASO-NUS/ASO-SSJ, and ASO-EX8/ASO-SSJ.

### 2.3. RNA Extraction and cDNA Synthesis

Forty-eight hours after ASO transfection, total RNA was isolated using Sepasol-RNA I reagent (Nacalai Tesque), in accordance with the manufacturer’s protocol. Total RNA was then treated with DNaseI Amplification Grade (Invitrogen, Carlsbad, CA), in accordance with the manufacturer’s protocol. Freshly prepared RNA was immediately used for cDNA synthesis to avoid degradation because the intron-retaining transcript was unstable. 

cDNA was synthesized at 55 °C for 30 min in a total volume of 20 μL containing 1 μg of total RNA, 60 μM random hexamer primers, 1 mM dNTPs, 50 mM Tris/HCl, 30 mM KCl, 8 mM MgCl_2_ pH 8.5, 20 U protector RNase inhibitor, and 10 U transcriptor reverse transcriptase (Roche Diagnostics GmbH, Mannheim, Germany).

### 2.4. Reverse-Transcription PCR (RT-PCR) Analysis

To amplify *SMN* transcripts, cDNA was amplified using a forward primer on exon 6 (Ex6F: 5′-TGG TAC ATG AGT GGC TAT CAT ACT-3′) and a reverse primer on exon 8 (Ex8R: 5′-GTG CTG CTC TAT GCC AGC ATT-3′) (Figure 1B). *Glyceraldehyde-3-phosphate dehydrogenase* (*GAPDH*) transcript was amplified as a reference gene transcript using the following primer set: GAPDH-F (5′-GAG TCA ACG GAT TTG GTC GT-3′) and GAPDH-R (5′-GAC AAG CTT CCC GTT CTC AG-3′) [32].

An aliquot of cDNA, equivalent to 100 ng of total RNA, was subjected to RT-PCR. The RT-PCR was performed in a reaction mixture with a total volume of 30 μL containing 1× PCR buffer, 2 mM MgCl_2_, 0.2 mM each dNTP, 0.3 μM each primer, and 1.0 U Fast Start Taq DNA Polymerase (Roche Applied Science, Mannheim, Germany). 

To determine the transcript levels, we performed semi-quantitative RT-PCR analysis followed by gel electrophoresis. The RT-PCR for the *SMN* transcript was performed as follows: initial denaturation at 94 °C for 7 min; 35 cycles of denaturation at 94 °C for 1 min, annealing at 56 °C for 1 min, and elongation at 72 °C for 1 min; and then final elongation at 72 °C for 7 min. Meanwhile, the RT-PCR for the *GAPDH* transcript was performed as follows: initial denaturation at 94 °C for 7 min; 25 cycles of denaturation at 94 °C for 1 min, annealing at 60 °C for 1 min, and elongation at 72 °C for 1 min; and final elongation at 72 °C for 7 min.

All amplicons were electrophoresed on 4% agarose gel and visualized by Midori Green staining (Nippon Genetics, Tokyo, Japan). The sizes of the RT-PCR products are shown in Figure 1B.

### 2.5. Subcloning and Sequence Analysis

The PCR product of *SMN* transcript was purified and cloned into a TA cloning vector, pGEM-T vector (Promega, Madison, WI, USA), before being transformed into competent *Escherichia coli* (Invitrogen, Carlsbad, CA) and propagated in LB solution. The transformed *E. coli* was grown on an LB plate containing IPTG and X-gal for blue/white colony selection.

The white colonies were selected and subjected to PCR using T7-SP6 primer set and were run on an agarose gel to select the target colonies based on the amplified product size. The PCR product corresponding to cryptic-exon (exon 7a)-containing transcripts was then purified using Nucleospin kit, in accordance with the manufacturer’s protocol (Takara Bio, Tokyo, Japan), and subjected to direct sequencing analysis. This sequencing analysis was outsourced to Fasmac Co. Ltd.

### 2.6. Statistics

To compare the splicing efficiencies among the ASOs, the software ImageJ (Version 2.1.0; National Institutes of Health, Bethesda, MD, USA) was used. All assays were carried out in triplicate and statistical analyses were performed using Microsoft Excel with the add-in software Statcel 3 (The Publisher OMS Ltd., Tokyo, Japan). Results reported as mean ± SD were analyzed by ANOVA with Tukey–Kramer post hoc test for comparisons between groups. **p* < 0.01 was considered statistically significant.

## 3. Results

### 3.1. Dose Effect of ASOs 

SMA type 1 fibroblasts were transfected with ASO-NUS, ASO-EX8, and ASO-SSJ. The target sites of these ASOs are shown in Figure 1A. ASO-NUS targeted the ISS-N1 site in *SMN2* intron 7 [15,16], ASO-EX8 targeted a putative exonic splicing enhancer (ESE) region including the SR protein 40-binding (SRp40-binding) site [33] in *SMN2* exon 8 (ESEfinder 3.0 [34]), and ASO-SSJ targeted the polypyrimidine tract (PolyPy in Figure 1A) and the AG dinucleotide at the 3’ splice site of *SMN2* exon 8 (AG in Figure 1A) [9]. Incidentally, ASO-SSJ blocks the binding sites of U2 auxiliary factor (U2AF) subunits. The 65-kDa subunit (U2AF65) contacts the polypyrimidine tract and the 35-kDa subunit (U2AF35) interacts with the AG dinucleotide [35]. Thus, ASO-SSJ may block the binding of U2AF subunits.

To analyze the effect of the dose of ASOs on the level of transcripts containing exon 7, each ASO was transfected into the SMA type 1 fibroblasts at three concentrations (50, 100, and 200 nM), followed by analysis of *SMN2* transcript expression (Figure 2A). Here, distilled water without ASOs was used as a reference. The intensity of each band was determined to evaluate the levels of transcript isoforms (Appendix A).

#### 3.1.1. ASO-NUS 

Exon 7 inclusion: All concentrations of ASO-NUS used in this study increased the level of exon-7-containing transcripts (Ex6/Ex7/Ex8) while decreasing the level of exon-7-lacking transcripts (Ex6/Ex8) (Figure 2A). This means that all ASO-NUS concentrations corrected the *SMN2* exon 7 splicing in SMA fibroblasts. Among the three concentrations, 100 nM ASO-NUS corrected the splicing most effectively (Ex6/Ex7/Ex8 in Figure 2A,B). The ratio of full-length transcript/delta-7 transcript (RFD) was the highest in cells treated with 100 nM ASO-NUS. The mean RFD values of ASO-NUS concentrations arranged in ascending order were as follows: 1.26 with 50 nM, 2.36 with 200 nM, and 6.87 with 100 nM (Figure 2B and Appendix A). The RFD values with 50 nM and 200 nM ASO-NUS were much lower than that with 100 nM ASO-NUS (*p* < 0.01).

Cryptic exon creation: In addition, 200 nM ASO-NUS produced a new transcript (an extra band of Ex6/Ex7a/Ex7/Ex8 in Figure 2A). Nucleotide sequencing analysis demonstrated that this transcript included a sequence of exon 7a between exons 6 and 7, which we previously reported as a cryptic exon in intron 6 [32] (Figure 3). The cryptic-exon-containing transcript was observed only in the cells treated with 200 nM ASO-NUS, but not in the cells treated with 50 and 100 nM ASO-NUS (Figure 2C).

Intron 7 retention: ASO-NUS at concentrations of 50, 100, and 200 nM produced no intron-7-containing transcripts (no bands of Ex6/Ex7/In7/Ex8 in Figure 2A).

#### 3.1.2. ASO-EX8 

Exon 7 inclusion: Regarding ASO-EX8, its use at concentrations of 50 and 100 nM corrected the *SMN2* exon 7 splicing in the SMA fibroblasts (Ex6/Ex7/Ex8 in Figure 2A). The RFD values with 50 and 100 nM ASO-EX8 were significantly higher than that with no ASOs (distilled water) (*p* < 0.01). However, they were much lower than with 100 nM ASO-NUS (*p* < 0.01) (Figure 2B). 

Cryptic exon creation: ASO-EX8 at 200 nM slightly corrected the *SMN2* exon 7 splicing, but clearly produced exon-7a-containing transcripts (Figure 2A,C, and Appendix A). 

Intron 7 retention: ASO-EX8 at 200 nM produced a significant amount of intron-7-retaining transcript (a clear band of Ex6/Ex7/In7/Ex8 in Figure 2A), but at concentrations of 50 and 100 nM it produced only a trace amount of this transcript (only faint bands of Ex6/Ex7/In7/Ex8 in Figure 2A).

#### 3.1.3. ASO-SSJ

Exon 7 inclusion: Regarding ASO-SSJ, its use at a concentration of 50 nM corrected the *SMN2* exon 7 splicing in SMA fibroblasts (Ex6/Ex7/Ex8 in Figure 2A). RFD values with 50 nM ASO-SSJ were significantly higher than those with no ASOs (distilled water) (*p* < 0.01). However, they were significantly lower than with 100 nM ASO-NUS (*p* < 0.01) (Figure 2B, Appendix A). 

Cryptic exon creation: ASO-SSJ at 200 nM did not correct the *SMN2* exon 7 splicing and clearly produced exon-7a-containing transcript (Figure 2A,C). 

Intron 7 retention: ASO-SSJ at concentrations of 50, 100, and 200 nM produced only trace amounts of intron-7-retaining transcript (only faint bands of Ex6/Ex7/In7/Ex8 in Figure 2A).

### 3.2. Effects of ASOs in Combination

In this experiment, combinations of two ASOs, each at a concentration of 100 nM (total 200 nM), were transfected into SMA type 1 fibroblasts, which were then analyzed for *SMN2* transcript expression after 48 h of incubation at 37 °C. The ASO combinations applied were ASO-NUS/ASO-EX8, ASO-NUS/ASO-SSJ, and ASO-EX8/ASO-SSJ. Single ASOs (100 nM) and distilled water without ASOs were used as references. The intensity of each band was determined to evaluate the levels of transcript isoforms (Appendix A). 

Cells transfected with 100 nM ASO-NUS had the highest RFD among all groups (*p* < 0.01), which was consistent with the results of the dose effect analysis. The mean RFD values from ASOs applied in combination arranged in ascending order were as follows: 0.25 with ASO-Ex8/ASO-SSJ (100 nM each), 1.29 with ASO-NUS/ASO-SSJ (100 nM each), and 2.07 with ASO-NS/ASO-EX8 (100 nM each) (*p* < 0.01) (Figure 4B and Appendix A).

#### 3.2.1. ASO-NUS/ASO-EX8

Exon 7 inclusion: The combination of ASO-NUS/ASO-EX8 (100 nM each) corrected the *SMN2* exon 7 splicing to some degree (Figure 4B) but showed significantly lower RFD values than the use of ASO-NUS alone (Appendix A). The presence of ASO-EX8 somewhat canceled out the effect of ASO-NUS (Figure 4B, Appendix A). 

Cryptic exon creation: It should be noted that the above combination of ASO-NUS/ASO-EX8 (100 nM each) was associated with high production of exon-7a-containing transcript (Ex6/Ex7a/Ex7/Ex8), as shown in Figure 4A,C. However, in the additional experiments with ASO-NUS/ASO-EX8, the cryptic exon was not activated when the total of combined dose was 100 nM, regardless of combination (or mixing) ratios (Appendix A). Thus, the excess of total amount of ASOs might be a factor to produce the cryptic-exon-containing transcript.

Intron 7 retention: ASO-NUS/ASO-EX8 (100 nM each) produced only a trace amount of intron-7-retaining transcript (a faint band of Ex6/Ex7/In7/Ex8 in Figure 4A).

#### 3.2.2. ASO-NUS/ASO-SSJ

Exon 7 inclusion: The combinations of ASO-NUS/ASO-SSJ (100 nM each) corrected the *SMN2* exon 7 splicing to some degree (Figure 4B) but were associated with significantly lower RFD values than the use of ASO-NUS alone (Appendix A). The presence of ASO-SSJ, as well as ASO-EX8, canceled out the effect of ASO-NUS. 

Cryptic exon creation: Notably, the combination of ASO-NUS/ASO-SSJ (100 nM each) produced a high level of the exon-7a-containing transcript (Ex6/Ex7a/Ex7/Ex8), as shown in Figure 4A,C. The cryptic exon activation was not clearly observed when the total dose of combined ASO-NUS/ASO-SSJ was 100 nM, regardless of mixing ratio (Appendix A). This observation was similar with the ASO-NUS/ASO-EX8 combination when the total dose was 100 nM. These observations suggested that the total amount of ASOs might be more critical than combination ratios. 

Intron 7 retention: The combination of ASO-NUS/ASO-SSJ (100 nM each) produced only a trace amount of intron-7-retaining transcript (a faint band of Ex6/Ex7/In7/Ex8 in Figure 4A).

#### 3.2.3. ASO-EX8/ASO-SSJ

Exon 7 inclusion: ASO-EX8/ASO-SSJ (100 nM each) had a lower effect on correcting *SMN2* exon 7 splicing than other combinations of ASO-NUS/ASO-EX8 or ASO-SSJ (100 nM each). The mean RFD value of ASO-Ex8/ASO-SSJ (100 nM each) was the same as that of cells treated with distilled water (with no ASO).

Cryptic exon creation: Moreover, the combination of ASO-EX8/ASO-SSJ (100 nM each) led to almost no production of the exon-7a-containing transcript (Figure 4A,C). Taken together with the results of other combinations, the activation of cryptic splice sites in intron 6 may be closely related to the presence of ASO-NUS and/or the total amount of ASOs.

Intron 7 retention: However, the combination of ASO-EX8/ASO-SSJ (100 nM each) produced a significant amount of the intron-7-retaining transcript (Ex6/Ex7/In7/Ex8 in Figure 4A).

## 4. Discussion

### 4.1. ASO Targeting ISS-N1 Site

Many ASOs for treating SMA have been studied, such as ASOs inhibiting alternative 3′ splice site pairing of *SMN2* exon 8 [9], ASOs targeting an intronic splicing suppressor site in *SMN2* intron 6 [10], peptide nucleic acid (PNA) with an arginine-serine (RS) domain that is a site for binding to exon 7 (known as the ESSENCE method) [11], ASOs containing a sequence complementary to exon 7 and a sequence non-complementary to some ESE motifs (known as the TOES method) [12,13], trans-splicing RNA carrying the exon 7 sequence [14], ASOs targeting the ISS-N1 site [15,16], and ASOs targeting the 5′ region of exon 8 [17]. 

As mentioned in the Introduction section, an ASO targeting ISS-N1, nusinersen, was approved as the first drug for SMA by the FDA. The ISS-N1 site was discovered by Singh et al. [36]. They were also the first to describe that deletion of this site restored inclusion of *SMN2* exon 7 [15]. In addition, in 2008 Hua et al. demonstrated that heterogeneous nuclear ribonucleoprotein (hnRNP) A1 and hnRNP A2 bind to the ISS-N1 site and that an ASO masking the motifs in the ISS-N1 site fully restored inclusion of *SMN2* exon 7 [16]. Subsequently, Ionis Pharmaceuticals (formerly ISIS Pharmaceuticals) began the clinical development of nusinersen, an antisense drug targeting the ISS-N1 site [37]. Nusinersen showed very promising results at all stages of clinical development and was approved by the FDA in 2016 [37].

Our study confirmed the findings in the previous studies reported by Singh et al. and Hua et al. [15,16]. According to our results, 100 nM ASO-NUS almost fully restored the inclusion of *SMN2* exon 7. However, 200 nM ASO-NUS restored this inclusion with lower efficiency than 100 nM ASO-NUS.

In addition, 200 nM ASO-NUS produced a cryptic exon (exon 7a) transcript in our experiment. In line with this, Ottesen et al. reported that a high concentration of an ASO targeting ISS-N1 caused massive perturbation of the transcriptome, including activation of a cryptic splice site in intron 6 [29]. Remarkably, combinations of ASOs, ASO-NUS/ASO-EX8 and ASO-NUS/ASO-SSJ, as well as a high dose of ASO-NUS, also activated the cryptic splice sites in intron 6 (Figure 4A). We discuss below the activation of the cryptic exon in *SMN2* intron 6.

### 4.2. ASO Targeting SMN2 Exon 8

In 2001, Lim et al. reported that an ASO directed toward the 3′ splice site of *SMN2* exon 8 altered *SMN2* splicing in favor of exon 7 inclusion [9]. This was surprising, as it would be expected that disruption of 3′ splice site function of exon 8 may cause intron 7 to be retained. Actually, they indeed showed that the level of intron 7 retention was slightly increased. However, the inclusion of exon 7 into the mRNA was clearly improved. 

In 2018, Flynn et al. tested the possibility of developing a strategy to retain *SMN2* intron 7 [17]. Because the authentic stop codon is located within exon 7, the intron 7-retaining transcript sequence should encode the full-length SMN protein. Flynn et al. used ASOs targeting not only the 3′ splice site but also the ESE motifs (e.g., hTra2beta-1 binding motif, SF2/ASF binding motif, SRp 40 binding motif) in exon 8. According to them, the ASOs increased the level of intron 7-retaining *SMN2* transcript, but this was not linked to an increase in the level of SMN protein because of the function of the longer 3’UTR in the intron 7-retaining transcript [17].

The results of Lim et al. and Flynn et al. were re-tested in our study [9,17]. According to our results, ASO-SSJ and ASO-EX8 slightly increased the level of transcripts with intron 7 retention (Ex6/Ex7/In7/Ex8 in Figure 2A and Figure 4A) and further enhanced the level of transcripts with exon 7 inclusion (Ex6/Ex7/Ex8 in Figure 2A and Figure 4A). ASOs targeting the 3′ splice site/5′ region of exon 8 may improve exon 7 inclusion, suggesting the existence of novel mechanisms potentially associated with unexpected exon 7 splicing. 

Pao et al. previously reported a synergistic effect of two ASOs [30]. According to them, the efficiency of exon 7 inclusion by two ASOs masking the ISS-N1 site and the 3’ splice site of exon 8 was much higher than that by a single ASO masking the ISS-N1 site [30]. However, our data did not show such a synergistic effect of a combination of ASOs. In our study, the combination of ASO-EX8/ASO-SSJ (100 nM each) was not associated with exon 7 inclusion, while the combination of ASO-NUS/ASO-EX8 (100 nM each) or ASO-NUS/ASO-SSJ (100 nM each) restored exon 7 inclusion to some degree, albeit with limited efficiency. The efficiency of exon 7 inclusion by a single ASO masking the ISS-N1 site (ASO-NUS) was much higher than that of two ASOs masking the ISS-N1 site and the 3′ splice site/5′ region of exon 8 (ASO-SSJ or ASO-EX8). Our results suggested that ASO-EX8 and ASO-SSJ may inhibit the ability of ASO-NUS to include exon 7.

### 4.3. Activation of Cryptic Splice Sites in SMN2 Intron 6 by ASOs

Our study demonstrated that a high dose (200 nM) of ASO-NUS, ASO-EX8, and ASO-SSJ activated cryptic splice sites in *SMN2* intron 6, leading to a cryptic exon, exon 7a. Ottesen et al. previously reported that a high concentration of ASOs targeting ISS-N1 activated cryptic splice sites in *SMN2* intron 6, leading to a cryptic exon, and considered it an off-target effect of ASOs targeting ISS-N1 [29]. 

*SMN* exon 7a is a cryptic exon that has been described independently by our group and Singh’s group (according to their nomenclature, it is exon 6B) [32,38]. The cryptic splice sites in intron 6 can be activated in both *SMN* genes, *SMN1* and *SMN2*. Exon 7a harbors two premature stop codons and gene products containing it may be subject to nonsense-mediated decay (NMD) [32]. If an exon-7a-containing transcript were translated into a truncated protein by a particular regulatory system, the product could have some functional activity and intermediate stability [38]. Even so, the inclusion of exon 7a in *SMN2* mRNA should be considered undesirable for the treatment of SMA with ASOs. 

In this study, we also observed that the combination of ASOs targeting ISS-N1 and the 3′ splice site of *SMN2* exon 8 activated the cryptic splice sites in intron 6, even though the concentration of each ASO was only 100 nM. A high concentration of single ASOs or the simultaneous use of two ASOs may alter the dynamic secondary structure of *SMN2* pre-mRNA, activating cryptic splice sites in intron 6. In other words, exon 7 splicing mechanisms are so delicate that some alteration of the dynamic secondary structure of pre-mRNA can easily affect the splicing pattern [39].

### 4.4. Research Limitations

We did not perform any protein and functional analyses of SMN in this study because we focused on the alternative splicing patterns caused by the administration of high dose of ASOs and combination of ASOs. However, we expected that the increase of full-length *SMN2* transcript led to the increase of functional full-length SMN protein in the cells and nuclear gems based on the report of Hua et al. [40], and the cryptic-exon-containing transcript could produce a truncated protein with some degree of functional activity and intermediate stability based on the report of Seo et al. [38]. 

## 5. Conclusions

Nusinersen, an ASO targeting the ISS-N1 site, is effective for improving the motor function of infants with SMA, but not so effective in advanced-stage patients. In addition, a ceiling effect of the current regimen has also been reported. To overcome these problems, studies have explored the effects of a high dose of nusinersen or the synergistic effects of combinations of different drugs. Based on our study, the RFD value with 100 nM ASO-NUS (an ASO targeting ISS-N1) was significantly higher than that with 50 nM ASO-NUS, suggesting a beneficial dose-dependent effect on exon 7 inclusion. However, 200 nM ASO-NUS showed significantly lower RFD values than 100 nM ASO-NUS and activated cryptic splicing sites. Above a certain threshold, the higher the concentration of ASO-NUS, the less effective and more harmful it may be. It should be noted that cryptic exon activation is linked to the decrease of full-length *SMN2* transcript, leading to the reduction of functional full-length SMN protein. In addition, combinations of ASO-NUS and other ASOs produced similar results to a high concentration of ASO-NUS. 

In conclusion, although it is difficult to directly apply the results of cultured cell studies to the clinical setting, our data suggested that a high concentration of ASOs or their use in combination may show unanticipated effects, including a lower rate of splicing correction and the creation of a cryptic exon. 

## Figures and Tables

**Figure 1 genes-13-00685-f001:**
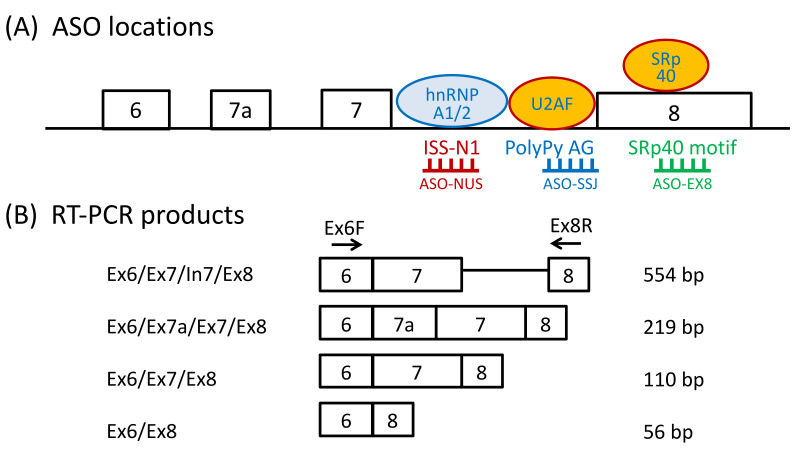
The locations of antisense oligonucleotides (ASOs) and reverse transcription-polymerase chain reaction (RT-PCR) products. (**A**) The targeting sites of ASOs (red, blue, and green lines) used in this study and their respective binding motifs. The numbered boxes and black lines represent the *survival motor neuron 2 (SMN2)* exons and introns, respectively. (**B**) The primers’ locations and RT-PCR products. The numbered boxes and black lines represent *SMN2* exons and introns, respectively. The sizes and composition of the transcript products are stated on the right and left of the figure. Abbreviations: hnRNPA1/2 (Heterogeneous nuclear ribonucleoprotein A 1/2); U2AF (U2 auxiliary factor); SRp40 (SR Protein 40); ISS-N1 (Intronic splicing silencer N1); PolyPy (polypyrimidine tract); AG (AG dinucleotide at 3’ splice site).

**Figure 2 genes-13-00685-f002:**
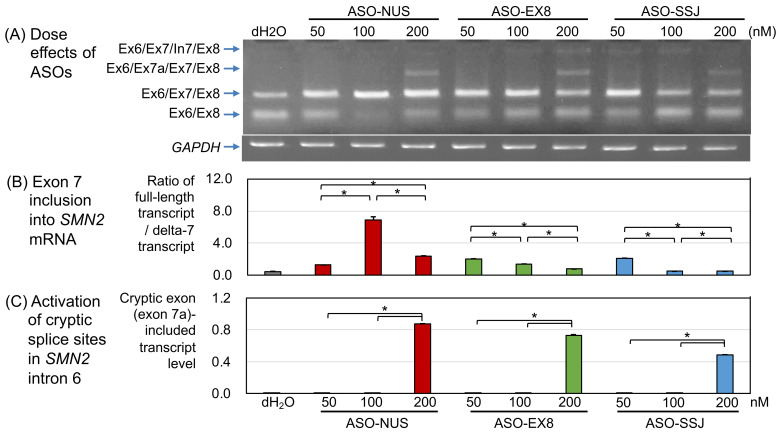
Dose effects of ASOs in spinal muscular atrophy (SMA) fibroblast. (**A**) Agarose gel electrophoresis of RT-PCR analysis. Lane 1 is control cells transfected with dH_2_O. The types and concentrations of ASOs used in this study are indicated here. The arrows indicate the transcript product of *SMN2* (upper panel) or *glyceraldehyde-3-phosphate dehydrogenase* (*GAPDH*) (lower panel). (**B**) Quantification of exon 7 inclusion into *SMN2* mRNA determined from the ratios of full-length transcript/delta-7 transcript (RFD). (**C**) Quantification of exon 7a into *SMN2* mRNA. An asterisk (*) indicates *p* < 0.01 in ANOVA.

**Figure 3 genes-13-00685-f003:**
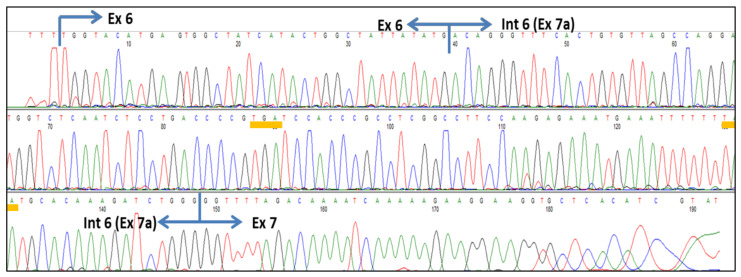
Partial nucleotide sequence of *SMN2* exon 6 to exon 7 showing the activated cryptic exon in intron 6. Blue arrows indicate the exon border. Yellow underlines indicate premature stop codons.

**Figure 4 genes-13-00685-f004:**
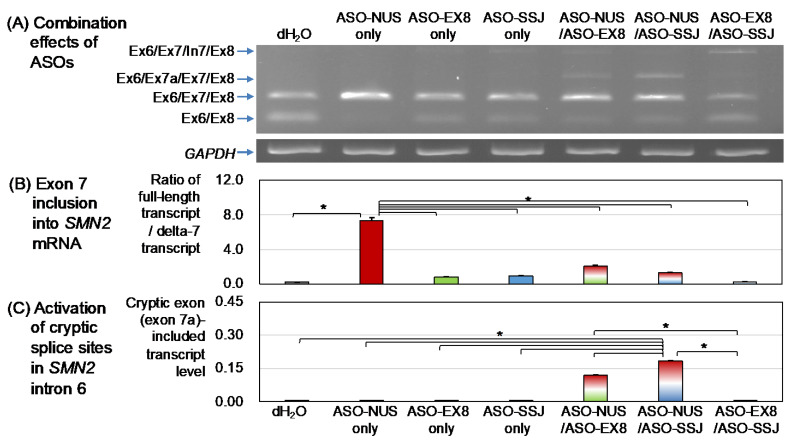
The effects of ASOs combination in SMA fibroblast. (**A**) Agarose gel electrophoresis of RT-PCR analysis. Combinations of two ASOs, each at a concentration of 100 nM (total 200 nM), were used in this study. Lane 1 is control cells transfected with dH_2_O. Lanes 2-4 are control cells with a single ASO (100 nM). The arrows indicate the transcript product of *SMN2* (upper panel) or *GAPDH* (lower panel). (**B**) Quantification of exon 7 inclusion into *SMN2* mRNA determined from the RFD. (**C**) Quantification of exon 7a into *SMN2* mRNA. An asterisk (*) indicates *p* < 0.01 in ANOVA.

## Data Availability

Not applicable.

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
