# Peer review of "High Concentration or Combined Treatment of Antisense Oligonucleotides for Spinal Muscular Atrophy Perturbed *SMN2* Splicing in Patient Fibroblasts"

_genes, 2022, doi:10.3390/genes13040685_

Round 1

Reviewer 1 Report

This manuscript by Wijaya et al., analyzed dose-dependency and combination-dependency of ASO on the correction of SMN2 splicing SMA cells. They showed that all ASOs are more efficient at lower concentration (50 or 100 nM) in splicing correction than high concentration (200 nM). They showed that a high concentration ASOs produced a cryptic exon in exon 6. a mixture of two types of ASOs (100 nM each) also produced cryptic exon. They then conclude that high concentration or combination of two ASOs perturb correcting roles on SMN2 splicing. This is a short manuscript, contents are not sufficient.

  1. In The title “High Concentration or Combined Treatment of Antisense Oli gonucleotides for Spinal Muscular Atrophy May Perturb SMN2 3 Splicing 4”, they used “May”, does this mean that their results are not coclusive?
  2. The authors provided only two gel pictures (figure 2) in the manuscript. Other figures are quantitation results of figure 2.
  3. They claimed that 100 nM concentration of two ASO also produce cryptic splicing. Is it because total concentration of Oligos are 200 nM?
  4. Figure 2, 3 and 4 can combine into in figure.

Author Response

Reviewer 1
General comment: This manuscript by Wijaya et al., analyzed dose-dependency and combinationdependency of ASO on the correction of SMN2 splicing SMA cells. They showed that all ASOs are more
efficient at lower concentration (50 or 100 nM) in splicing correction than high concentration (200 nM). They
showed that a high concentration ASOs produced a cryptic exon in exon 6. a mixture of two types of ASOs
(100 nM each) also produced cryptic exon. They then conclude that high concentration or combination of
two ASOs perturb correcting roles on SMN2 splicing. This is a short manuscript; contents are not sufficient.
Answer: We thank you very much for your careful reading and helpful comments. In our small study, we
showed a high concentration of ASOs produced a cryptic exon in exon 6 and a mixture of two types of
ASOs also produced a cryptic exon, as you summarized properly. However, at the same time, we
demonstrated that activation of the cryptic exon linked to the production of less functional SMN protein
and reduction of fully functional SMN protein, which may be a serious condition in the current treatment
for SMA patients with ASOs. Thus, we think it is necessary to report our results as soon as possible.
1. In The title “High Concentration or Combined Treatment of Antisense Oligonucleotides for Spinal
Muscular Atrophy May Perturb SMN2 Splicing”, they used “May”, does this mean that their results are
not coclusive?
Answer: We thank you very much for your constructive feedback.
We obtained some results using cultured cells treated with high concentrations of ASOs; and the results
suggested the possible perturbation in the SMN2 splicing patterns. The concentrations of ASOs in this
study may be higher than those of therapeutic concentration in SMA patients treated with nusinersen.
The concentrations used in our study were 50-200 nM. These concentrations had been used in the study of
Ottesen et al. (reference number 29). But it is difficult to directly compare the results of cultured cell studies
and clinical studies. According to MacCannel et al (2021), the trough concentrations of nusinersen were
approximately 10 ng/mL (1.33 nM). However, the maximum cerebrospinal fluid (CSF) concentrations must
be much higher than C trough concentrations. The CSF concentration is changing during the treatment
course. In addition, we did not know the differences in the responses against ASOs between neuronal cells
and fibroblasts. Therefore, we used the word "may" in the title“High Concentration or Combined Treatment
of Antisense Oligonucleotides for Spinal Muscular Atrophy May Perturb SMN2 Splicing”.
Even so, our results indicated the possibility that high dose of ASO may cause perturbation in the SMN2
splicing patterns, although we could not exactly determine what concentration of ASO should be critical
in the clinical settings. Here, following your suggestion, we omit the word “may” in the title, but added "
in SMA fibroblasts” at the end of the title.
The new title in the revised version is “High Concentration or Combined Treatment of Antisense
Oligonucleotides for Spinal Muscular Atrophy Perturbed SMN2 Splicing in Patient Fibroblasts”.
We also mentioned the difficulty of direct comparison between studies using cultured cells and clinical
settings in the conclusion section.
page 11, 5. Conclusion (6. Conclusion in the revised version)
Original version (Last sentence) In conclusion, our data suggested that a high concentration of
ASOs or their use in combination may show unanticipated effects, including a lower rate of splicing
correction and the creation of a cryptic exon.
Revised version (Last sentence) In conclusion, although it is difficult to directly apply the results
of cultured cell studies to the clinical setting, our data suggested that a high concentration of ASOs
or their use in combination may show unanticipated effects, including a lower rate of splicing
correction and the creation of a cryptic exon.
2. The authors provided only two gel pictures (figure 2) in the manuscript. Other figures are quantitation
results of figure 2.
Answer: We thank you for your comment. This study focusing on the splicing alternative caused by the
high dose and combination of two antisense thus the figure focus on the transcript analyses and the
quantitation.
Following your suggestion in point 4, we combined the transcript analyses and the quantitation into one
figure.
3.They claimed that 100 nM concentration of two ASO also produce cryptic splicing. Is it because total
concentration of Oligos are 200 nM?
Answer: We thank you for your comment. We observed that cryptic splicing occurred when the total of
combined dose was 200nM similar with single antisense administration with a total dose of 200 nM. When
the total combined dose of two ASOs was less than 200 nM, the cryptic exon was not clearly observed (data
not shown). Thus, the total amount of ASOs might also be a factor to produce the cryptic exon-containing
transcript, as you suggested.
Following your suggestion, we added the description that the total amount of ASOs might be a factor to
produce the cryptic exon in the revised version.
3.2.1. ASO-NUS/ASO-EX8 [Cryptic exon creation]
We put the following sentence, "However, in the additional experiments with ASO-NUS/ASO-EX8,
the cryptic exon was not activated until the total of combined dose reached 200nM (data not shown).
Thus, the total amount of ASOs might be a factor to produce the cryptic exon-containing transcript."
3.2.2. ASO-NUS/ASO-SSJ [Cryptic exon creation]
We put the following sentence, "However, in the additional experiments with ASO-NUS/ASO-SSJ,
the cryptic exon was not activated until the total of combined dose reached 200nM (data not shown).
Thus, the total amount of ASOs might be a factor to produce the cryptic exon-containing transcript."
3.2.3 ASO-EX8/ASO-SSJ [Cryptic exon creation]
We revised the last sentence as follows, "Taken together with the results of other combinations, the
activation of cryptic splice sites in intron 6 may be closely related to the presence of ASO-NUS and/or
the total amount of ASOs."
4. Figure 2, 3 and 4 can combine into in figure.
Answer: We thank you for your suggestion. Following your suggestion about transcript analysis in the gel
and their quantitation, we combined the figures 2 A and 3 into one figure (new figure 2), and figures 2B
and 5 into one figure (new figure 3). We spared figure 4 because it is describing the sequence result.
5. The reviewer 1 said, “Extensive editing of English language and style required”. Here, we show the
certification of English-editing by a scientist and native English speaker

Please also see the attachment

Reviewer 2 Report

Comments to the editor and authors

There have been advances in the treatment options for spinal muscular atrophy (SMA). Antisense oligonucleotide (ASO)-based drug is emerging as a powerful tool for targeting and repairing mutated transcripts to parietally or fully restore their activities in cells. In this manuscript, Wijaya et al utilized published ASOs that are known to promote the correction of SMN2 exon 7 splicing and the production of the full-length SMN2 mRNA. The authors tested whether a higher concentration of individual ASO correlates with an increased amount of full-length SMN2 mRNA or whether two-ASO cocktails can result in a significant inclusion of exon 7 and amount of full-length SMN2 mRNA compared to individual ones. ASOs at a high concentration or in cocktails resulted in less production of the full-length SMN2 mRNA maybe because of the generation of a cryptic exon.  

Suggestions:

In the manuscript, the authors did not support their results for high concentration (200nM) or two-ASOs cocktails versus 50-100nM ASO by protein and functional studies.  

-The authors should show the intracellular levels of SMN protein in the transfected and untransfected fibroblasts using western blot analyses. 

- The authors can also show the formation of nuclear Gemini of coiled bodies (GEMs) using immunofluorescence and Nuclear GEM quantification. 

Author Response

Reviewer 2
Comments to the editor and authors
General comment: There have been advances in the treatment options for spinal muscular atrophy (SMA).
Antisense oligonucleotide (ASO)-based drug is emerging as a powerful tool for targeting and repairing mutated transcripts to parietally or fully restore their activities in cells. In this manuscript, Wijaya et al utilized published ASOs that are known to promote the correction of SMN2 exon 7 splicing and the
production of the full-length SMN2 mRNA. The authors tested whether a higher concentration of individual ASO correlates with an increased amount of full-length SMN2 mRNA or whether two-ASO cocktails can result in a significant inclusion of exon 7 and amount of full-length SMN2 mRNA compared
to individual ones. ASOs at a high concentration or in cocktails resulted in less production of the fulllength SMN2 mRNA maybe because of the generation of a cryptic exon.

Answer: We thank you very much for your careful reading and helpful comments. We carefully studied your comments and suggestions, we added some sentences and a new subsection, "research limitations".
Suggestions:
1. In the manuscript, the authors did not support their results for high concentration (200nM) or two-ASOs cocktails versus 50-100nM ASO by protein and functional studies.
Answer: We thank you for your valuable comments. We did not perform any protein and functional analyses in this study, because we focused on the alternative splicing patterns caused by the administration of high dose of ASOs and combination of ASOs.
However, we are able to expect that activated cryptic exon can produce a truncated protein with some degree of functional activity and intermediate stability based on the report of Seo et al. (reference number 38). Although the truncated protein is somewhat functional, the inclusion of exon 7a should be considered undesirable effect, because cryptic exon activation is linked to the decrease of full-length SMN2 mRNA,
leading to the reduction of functional full-length SMN protein.
Following your suggestion, we added a sentence in the conclusion of the revised version.
6. Conclusion
We put the following sentence, "It should be noted that cryptic exon activation is linked to the decrease of full-length SMN2 mRNA, leading to the reduction of functional full-length SMN protein."
2. The authors should show the intracellular levels of SMN protein in the transfected and untransfected fibroblasts using western blot analyses.
3. The authors can also show the formation of nuclear Gemini of coiled bodies (GEMs) using immunofluorescence and Nuclear GEM quantification.
Answer: We thank you for your valuable comments. As we mentioned above, we did not perform any protein analyses including Western blotting or nuclear GEM quantification, because we focused on the
alternative splicing patterns caused by the administration of high dose of ASOs and combination of ASOs.
However, we are able to expect that the increase of full-length SMN2 mRNA led to the increase of functional full-length SMN protein and nuclear GEM numbers in the cells, according to the report of Hua (reference number 40).
Following your suggestion, we added a new subsection of research limitations in the revised version.
5. Research limitations
We did not perform any protein and functional analyses of SMN in this study, because we focused on the alternative splicing patterns caused by the administration of high dose of ASOs and combination of ASOs. However, we expected that the increase of full-length SMN2 transcript led to the increase of functional full-length SMN protein in the cells and nuclear gems (nuclear Gemini of coiled bodies) based on the report of Hua et al. [40], and the cryptic exon-containing transcript could produce a truncated protein with some degree of functional activity and intermediate stability based on the report of Seo et al. [38]. 

Round 2

Reviewer 1 Report

Although the authors revised the manuscript and added some information, the manuscript does not include sufficient contents to support publications in "genes".

Reviewer 2 Report

Thanks for the changes the authors made to the revised version. I think I would accept the focus on mRNA splicing since the limitation is clarified. 

Good luck
